# CRISPR/Cas9 mutagenesis invalidates a putative cancer dependency targeted in on-going clinical trials

Ann Lin[1,2†], Christopher J Giuliano[1,2†], Nicole M Sayles[1], Jason M Sheltzer[1*]

[1]Cold Spring Harbor Laboratory, Cold Spring Harbor, United States; [2]Stony Brook University, Stony Brook, United States

**Abstract** The Maternal Embryonic Leucine Zipper Kinase (MELK) has been reported to be a genetic dependency in several cancer types. MELK RNAi and small-molecule inhibitors of MELK block the proliferation of various cancer cell lines, and MELK knockdown has been described as particularly effective against the highly-aggressive basal/triple-negative subtype of breast cancer. Based on these preclinical results, the MELK inhibitor OTS167 is currently being tested as a novel chemotherapy agent in several clinical trials. Here, we report that mutagenizing MELK with CRISPR/Cas9 has no effect on the fitness of basal breast cancer cell lines or cell lines from six other cancer types. Cells that harbor null mutations in MELK exhibit wild-type doubling times, cytokinesis, and anchorage-independent growth. Furthermore, MELK-knockout lines remain sensitive to OTS167, suggesting that this drug blocks cell division through an off-target mechanism. In total, our results undermine the rationale for a series of current clinical trials and provide an experimental approach for the use of CRISPR/Cas9 in preclinical target validation that can be broadly applied.

**\*For correspondence:** sheltzer@
cshl.edu

[†]These authors contributed
equally to this work

**Competing interests:** The
authors declare that no
competing interests exist.

**Reviewing editor:** Jeffrey
Settleman, Calico Life Sciences,
United States

## Introduction

Tumors of the breast can be divided into five distinct subtypes based on characteristic gene expression patterns. These breast cancer subtypes are referred to as Luminal A, Luminal B, Her2-enriched, normal-like, and basal (*Sørlie et al., 2001*). Basal breast cancers (BBCs) comprise ~15% of all diagnosed breast cancers and express genes typically found in the basal/myoepithelial layer of the mammary gland (*Badve et al., 2011*; *Rakha et al., 2008*). BBCs are most frequently diagnosed in younger patients and present with advanced histologic grade, central necrosis, and high mitotic activity. Additionally, ~70% of basal breast cancers fail to express the estrogen receptor (ER), the progesterone receptor (PR), or the human epidermal growth factor receptor 2 (HER2) (*Badve et al., 2011*). Tumors that lack expression of ER, PR, or HER2 are referred to as 'triple-negative' breast cancers, and are irresponsive to hormonal or anti-HER2 therapies that have proven effective against receptor-positive cancers. Due to their resistance to targeted therapies as well as their rapid rate of cell division, basal breast cancers currently have the worst prognosis of any breast cancer subtype. Thus, there is an urgent need to develop new therapies that are effective against triple-negative or basal-type tumors.

In recent years, significant progress has been made in the treatment of certain malignancies by targeting cancer cell 'addictions', or genetic dependencies that encode proteins required for the growth of specific cancer types (*Luo et al., 2009*). Drugs that block the function of a cancer dependency – like the antibody Herceptin in Her2+ breast cancer – can trigger apoptosis and durable tumor regression (*Weinstein, 2002*). Cancer cell addictions are often investigated through the use of different transgenic technologies to disrupt the expression of a specified gene. Two of the most

**eLife digest** Like a person who is dependent on coffee to be productive, cancer cells are dependent on the products of certain genes in order to dominate their environment and grow. Cancer cells will stop growing and die when the activity of these gene products is blocked. These genes are known as cancer dependencies or "addictions". As a result, researchers are constantly looking for cancer dependencies and developing drugs to block their activity.

It was previously believed that a gene called MELK was an addiction in certain types of breast cancer. In fact, pharmaceutical companies had developed a drug to block the activity of MELK, and this drug is currently being tested in human patients. However, Lin, Giuliano et al. have now taken a second look at the role of MELK in breast cancer, and have come to a different conclusion.

Using a gene editing technology called CRISPR/Cas9, Lin, Giuliano et al. removed MELK activity from several cancer cell lines. This did not stop cancer cells from multiplying, suggesting that MELK is not actually a cancer addiction.

Additionally, when breast cancer cells that do not produce MELK were exposed to the drug that is supposed to block MELK activity, the drug still stopped cell growth. Since the drug works when MELK is not present in the cell, the drug must be binding to other proteins. This suggests that MELK is not the actual target of the drug.

Lin, Giuliano et al. suggest that, in the future, CRISPR/Cas9 technology could be used to better identify cancer dependencies and drug targets before cancer drugs are given to human patients.

popular methodologies are RNA interference, which destabilizes a targeted transcript, and CRISPR mutagenesis, which utilizes the nuclease Cas9 to induce frameshift mutations at a targeted locus. While CRISPR-mediated genetic engineering has been widely adopted since its discovery in 2013, RNA interference remains popular due to its ability to deplete multiple isoforms of a protein, its reversibility, and its relative insensitivity to gene copy number (*Boettcher and McManus, 2015*). Moreover, the partial loss-of-function phenotype generated by RNAi may more accurately recapitulate the effects of drug treatment than the complete loss-of-function phenotype generated by a Cas9-induced frameshift mutation. Nonetheless, RNAi constructs exhibit limited specificity, and off-target knockdowns are an inherent and widespread problem in RNAi experiments (*Jackson et al., 2003*, *2006*; *Singh et al., 2011*).

The Maternal Embryonic Leucine Zipper Kinase (MELK) has received substantial attention as a potential cancer cell addiction and promising target for drug development. MELK was first identified as an AMPK family member expressed in the mouse pre-implantation embryo (*Heyer et al., 1997*), and has since been implicated in several cellular processes, including apoptosis (*Jung et al., 2008*), splicing (*Vulsteke et al., 2004*), and neurogenesis (*Nakano et al., 2005*). MELK is also over-expressed in most types of solid tumors, including breast, colon, liver, lung, melanoma, and ovarian cancer (*Gray et al., 2005*). Furthermore, many publications have reported that knocking down MELK using RNAi inhibited the proliferation of cell lines derived from these cancer types (*Gray et al., 2005*; *Lin et al., 2007*; *Kuner et al., 2013*; *Du et al., 2014*; *Kig et al., 2013*; *Speers et al., 2016*; *Alachkar et al., 2014*; *Marie et al., 2008*; *Nakano et al., 2008*; *Hebbard et al., 2010*; *Wang et al., 2014*; *Choi et al., 2011*; *Xia et al., 2016*; *Gu et al., 2013*). In particular, MELK has been identified as a key driver of basal-type breast cancer, suggesting a novel therapeutic approach to treat this disease (*Wang et al., 2014*). In response to the widespread reports that MELK is a cancer dependency, several companies have developed small molecule inhibitors of MELK that block the activity of the kinase in vitro and that inhibit cancer cell proliferation at micromolar or nanomolar concentrations (*Beke et al., 2015*; *Touré et al., 2016*; *Johnson et al., 2015a*, *2015b*; *Chung et al., 2012*). Additionally, four clinical trials have been launched to test the MELK inhibitor OTS167 in human cancers (NCT01910545, NCT02768519, NCT02795520, and NCT02926690).

As part of a project in our lab to characterize genes whose expression is associated with patient prognosis in cancer (*Sheltzer, 2013*), we identified MELK as highly-expressed in deadly tumors from multiple cancer types (data not shown). We set out to use CRISPR/Cas9 to characterize the effects of

MELK loss on tumorigenesis. Unexpectedly, we found that mutating MELK failed to affect the growth of every cancer cell line that we tested. Furthermore, the MELK inhibitor OTS167 remained effective against cells with null mutations in MELK, suggesting that its in vivo activity results from an off-target effect. We propose that CRISPR represents an essential modality to confirm putative cancer dependencies and drug specificity before preclinical findings are advanced to human trials.

## Results

### Mutagenizing MELK using CRISPR/Cas9

Over a dozen previous publications have reported that MELK is a cancer dependency, as blocking MELK with RNAi or small molecules inhibited the proliferation of cell lines derived from multiple tumor types (*Gray et al., 2005*; *Lin et al., 2007*; *Kuner et al., 2013*; *Du et al., 2014*; *Kig et al., 2013*; *Speers et al., 2016*; *Alachkar et al., 2014*; *Marie et al., 2008*; *Nakano et al., 2008*; *Hebbard et al., 2010*; *Wang et al., 2014*; *Choi et al., 2011*; *Xia et al., 2016*; *Gu et al., 2013*). However, several discrepancies exist in the literature on MELK. For instance, various publications disagree over the cell cycle stage affected by MELK inhibition (*Du et al., 2014*; *Kig et al., 2013*; *Alachkar et al., 2014*; *Wang et al., 2014*; *Beke et al., 2015*), while other publications disagree over whether receptor-positive breast cancer cell lines are sensitive (*Lin et al., 2007*; *Beke et al., 2015*; *Chung et al., 2012*) or resistant (*Wang et al., 2014*) to MELK inhibition. To unambiguously determine the effects of MELK loss in cancer cell lines, we applied CRISPR/Cas9 to generate frameshift mutations in the MELK coding sequence. We designed seven guide RNAs (gRNAs) against MELK, five of which target the N-terminal kinase domain and two of which target the C-terminal kinase-associated domain (*Figure 1A* and *Supplementary file 1*). Then, we cloned each guide RNA into a GFP-expressing vector and transduced the guides into three Cas9-expressing cell lines: the triple-negative breast cancer cell lines Cal51 and MDA-MB-231, reported to be addicted to MELK expression (*Wang et al., 2014*), and the melanoma cell line A375, from a cancer type that over-expresses MELK (*Gray et al., 2005*; *Ryu et al., 2007*). As negative controls in these assays, we also cloned and transduced three gRNA's that target the non-essential and non-coding Rosa26 locus (*Supplementary file 1*).

To assess the efficacy of our CRISPR system, we purified GFP+ populations from cell lines harboring each individual guide RNA, and then we analyzed the targeted loci by Sanger sequencing. TIDE analysis, which decomposes raw sequencing traces into linear combinations of indel mutations (*Brinkman et al., 2014*), revealed high cutting efficiency at most targeted loci (*Figure 1B* and *Figure 1—figure supplements 1–3*). Across the 21 samples, the median level of indel formation was 80%. These values likely underestimate the true mutation frequency, as TIDE analysis is not able to detect missense mutations and large indels will not be efficiently amplified by PCR. Western blot analysis of MELK protein levels in A375, Cal51, and MDA-MB-231 sorted populations further confirmed that our CRISPR system effectively ablated MELK expression (*Figure 1C* and *Figure 1—figure supplement 4*).

We next set out to determine whether MELK was required for cancer cell fitness. To test this, we measured cell proliferation over 15 days in culture in the 30 independent lines of A375, Cal51, and MDA-MB-231 that we had generated (*Figure 1D–F*). Surprisingly, we failed to detect any difference in proliferative capacity between the cell lines with wild-type or mutant MELK. For instance, in the A375 cell line, we calculated a mean doubling time of 16.9 hr among cells transduced with Rosa26 guide RNAs, while the cell lines transduced with MELK gRNAs exhibited a mean doubling time of 16.8 hr. MELK gRNA-transduced cell lines also exhibited wild-type levels of growth under anchorage-independent conditions (*Figure 1G–I*). These results call into question the notion that MELK is a genetic dependency either across cancer types or in triple-negative breast cancers.

### MELK is not a common cancer cell dependency

In order to assess whether a wider range of cancer cell lines were dependent on MELK for viability, we performed individual GFP dropout experiments in 13 Cas9-expressing cancer cell lines (*Shi et al., 2015*). In these assays, cancer cells are transduced with GFP-expressing guide RNA vectors at low MOI to create mixed populations of GFP+ and GFP- cells. A guide RNA that induces mutations in a gene required for cancer cell fitness will drop out from the population, resulting in a

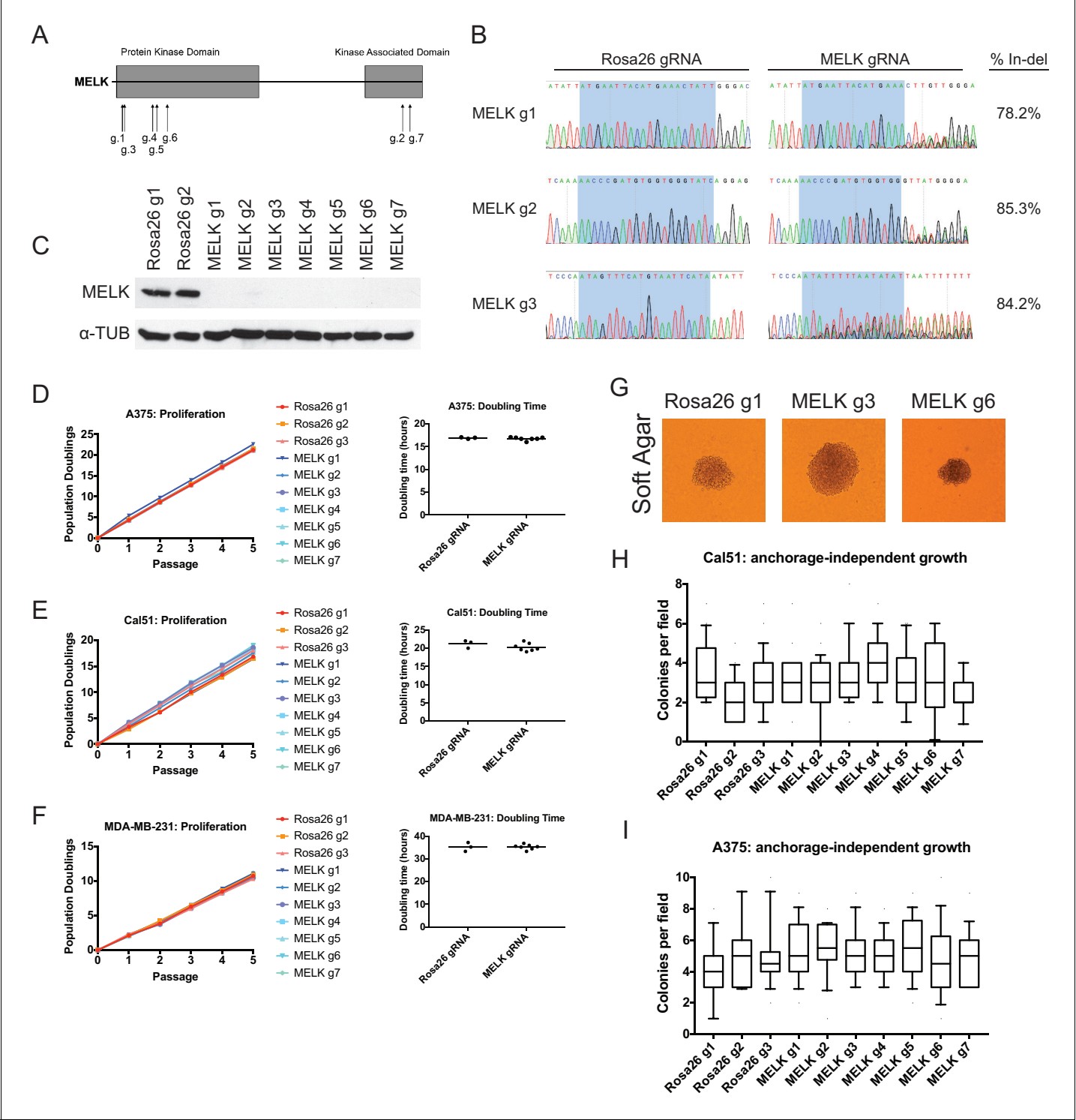

**Figure 1.** Mutation of the MELK kinase domain does not affect cancer cell proliferation or anchorage-independent growth. (**A**) Domain structure of MELK and locations of the sequences targeted by 7 MELK gRNAs. (**B**) Genomic DNA was purified from the indicated population of MDA-MB-231 cells and the targeted loci were amplified by PCR. Percent indel formation was estimated using TIDE analysis. The highlighted region indicates 20 nucleotides or 15 nucleotides of the sequence recognized by the guide RNA. Other sequence traces are presented in *Figure 1—figure supplements 1–3*. (**C**) Western blot analysis of GFP+ MDA-MB-231 cells using the Abcam ab108529 MELK antibody. Alpha-tubulin levels were analyzed as a loading control. (**D–F**) Proliferation and doubling time analysis of A375, Cal51, and MDA-MB-231 cell lines transduced with 3 Rosa26 gRNAs or with 7 MELK gRNAs. (**G**) Images of colonies from the indicated Cal51 strains grown in soft agar. (**H–I**) Quantification of anchorage-independent growth in Cal51 or

*Figure 1 continued on next page*

*Figure 1 continued*

A375 cells transduced with the indicated gRNA. For each assay, colonies were counted in at least 15 fields under a 10x objective. Boxes represent the 25th, 50th, and 75th percentiles of colonies per field, while the whiskers represent the 10th and 90th percentiles.

The following figure supplements are available for figure 1:

**Figure supplement 1.** Mutation of MELK using seven different guide RNAs in the A375 melanoma cell line.
**Figure supplement 2.** Mutation of MELK using seven different guide RNAs in the Cal51 triple-negative breast cancer cell line.
**Figure supplement 3.** Mutation of MELK using seven different guide RNAs in the MDA-MB-231 triple-negative breast cancer cell line.
**Figure supplement 4.** Western blot analysis of MELK-disrupted cell populations.

decreasing ratio of GFP+ to GFP- cells over time. Additionally, we considered it possible that cells had adapted to MELK loss during the time required to sort and expand pure GFP+ cell populations to perform the experiments described in *Figure 1*. For the following dropout assays, we monitored GFP levels directly following introduction of the gRNA virus, without selecting or expanding cell populations. Importantly, this strategy allows us to detect whether the mutation of MELK results in a transient or immediate loss of cell fitness.

As negative controls in this experiment, we utilized three gRNA's targeting Rosa26, and as positive controls we designed six gRNA's targeting the essential replication genes RPA3 and PCNA (*Supplementary file 1*). We first transduced these gRNA's individually into seven triple-negative breast cancer cell lines (Cal51, HCC1143, HCC1937, HCC70, MDA-MB-231, MDA-MB-453, and MDA-MB-468). Over the course of five passages in culture, gRNA's targeting Rosa26 typically depleted 1.2 to 2-fold (*Figure 2A*). This low level of depletion may result from off-target mutagenesis or from cell cycle arrest caused by repeated DNA breaks (*Aguirre et al., 2016*). Over the same period of time, gRNA's targeting RPA3 and PCNA depleted 5-fold to 100-fold. These positive control guides exhibited varying degrees of dropout (e.g., compare RPA3 g1 and RPA3 g2), which may result from variability in cutting efficiency or from functionally-important differences in the protein domains targeted by these guides. However, in every cell line tested, every single gRNA targeting RPA3 or PCNA dropped out to a greater degree than every single Rosa26 guide. In contrast to RPA3 and PCNA, the seven guides that targeted MELK typically depleted less than 2-fold. Across seven different gRNAs tested in seven different cell lines, we never observed a MELK guide deplete more than 2.5-fold. In six of the seven cell lines, a Rosa26 gRNA exhibited a higher level of depletion than every single MELK gRNA (*Figure 2B*). We conclude that these seven triple-negative breast cancer cell lines are not dependent on MELK for cell fitness.

To extend these observations, we repeated the GFP dropout experiments in six Cas9-expressing cell lines (A375, Cama1, HCT116, NCI-H1299, T24, and U118-MG) from other cancer types previously suggested to require MELK expression. Consistent with our observations in the triple-negative breast cancer cells, guides targeting Rosa26 exhibited minimal dropout over five passages, while guides targeting RPA3 or PCNA were depleted up to 90-fold (*Figure 2—figure supplement 1*). However, 0 of the 7 MELK guides exhibited significant dropout in any of the cell lines tested (maximum dropout: 1.8-fold). Again, guides targeting Rosa26 exhibited an equivalent or occasionally greater degree of depletion than guides targeting MELK (*Figure 2—figure supplement 1B*). In total, this data suggests that MELK is not a common cancer dependency.

## Unbiased RNAi and CRISPR screens fail to identify MELK as a cancer dependency

Several laboratories have conducted genome-wide or kinase-focused screens to identify novel cancer addictions. If these unbiased screens indicated that cancer cell lines required MELK expression to proliferate, then that would bolster the contention that MELK could be a therapeutic target in cancer. We therefore examined data from four recent screens: a kinome-wide siRNA screen in 117 cancer cell lines (*Campbell et al., 2016*), a genome-wide CRISPR screen in 6 cell lines (*Hart et al., 2015*), a genome-wide shRNA screen in 72 cancer cell lines (*Marcotte et al., 2012*; *Hart et al.,*

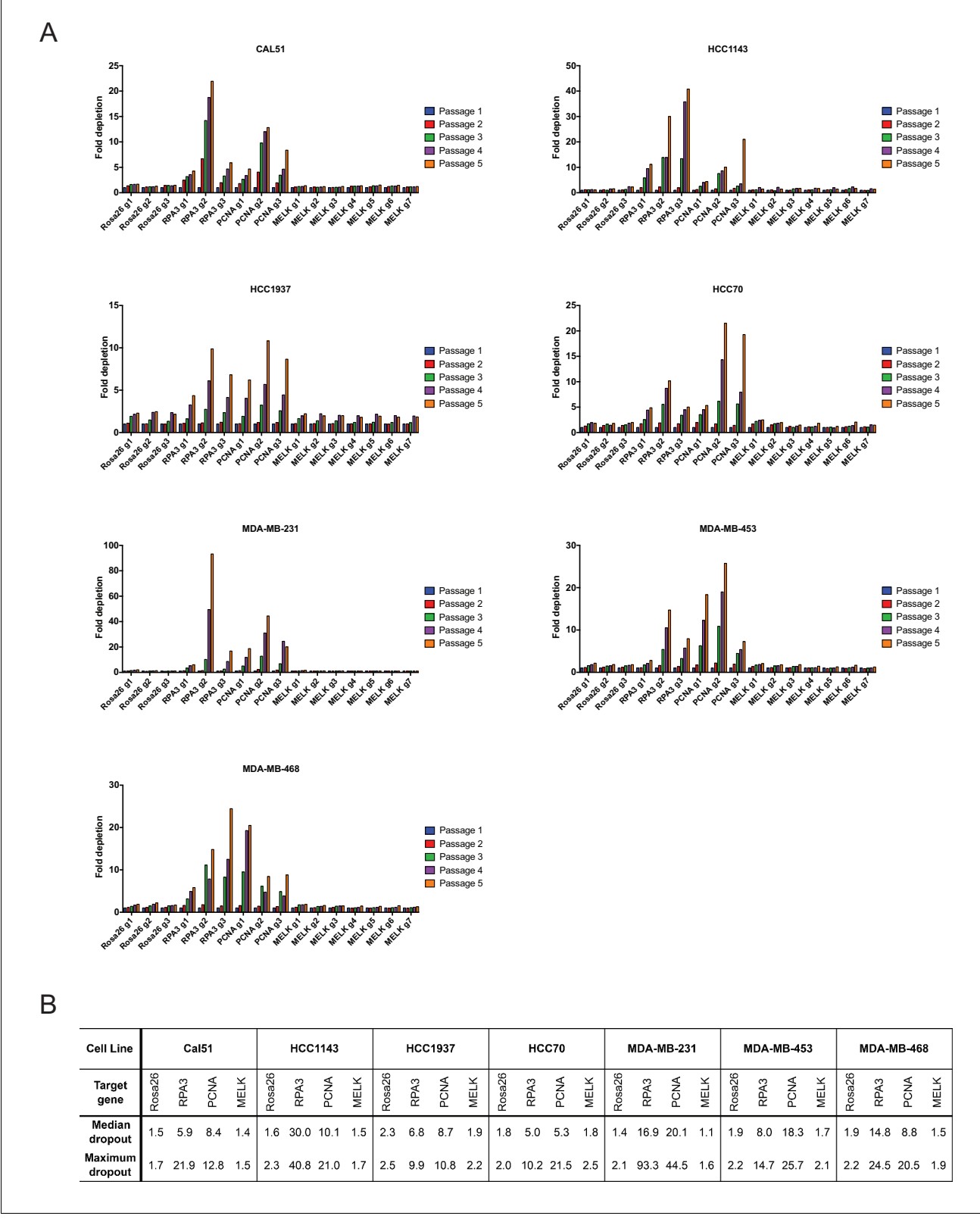

**Figure 2.** Guide RNAs targeting MELK fail to drop out in triple-negative breast cancer cell line competition experiments. (**A**) The fold change in the percentage of GFP+ cells, relative to the percentage of GFP+ cells at passage 1, is displayed for seven triple-negative breast cancer cell lines. (**B**) A table summarizing the results presented in (**A**) is displayed.

*Figure 2 continued on next page*

*Figure 2 continued*

The following figure supplements are available for figure 2:

**Figure supplement 1.** Guide RNAs targeting MELK fail to drop out in several cancer cell lines.

**Figure supplement 2.** Unbiased screens do not identify MELK as a cancer dependency.

---

*2014*), and a genome-wide CRISPR screen in seven cell lines (*Tzelepis et al., 2016*). Large-scale unbiased screens are prone to experimental artifacts, and variations in protocol, technology, or the method of analysis can cause different screens to yield different results (*Mohr et al., 2014*). Nonetheless, each of these screens identified multiple mitotic kinases as essential in various cancer cell lines, including Aurora B, BubR1, CDK1, and Plk1 (*Figure 2—figure supplement 2*). In contrast, MELK was not identified as essential in a single experiment. These negative results include 13 triple-negative breast cancer cell lines tested by Campbell et al. and 16 triple-negative breast cancer cell lines tested by the Moffatt lab. Several other published pan-cancer or breast cancer-focused screens have failed to identify MELK as either a general cancer dependency or a triple-negative breast cancer dependency (*Silva et al., 2008*; *Marcotte et al., 2016*; *Cowley et al., 2014*). Thus, while we do not consider results from large-scale screens to be dispositive, we believe that these findings, coupled with our own experimental evidence, suggest that MELK expression is not required for cancer cell proliferation.

## OTS167 inhibits the growth of receptor-positive breast cancer cell lines and cells that harbor mutant MELK

If MELK is not a cancer cell dependency, then drugs that inhibit MELK must either be ineffective at stopping cancer cell division or they must also act on other cellular targets. We therefore assessed the efficacy of the MELK inhibitor OTS167 (alternately called OTSSP167), a therapeutic agent being tested in several clinical trials. We treated a variety of cancer cell lines with 7-point serial dilutions of OTS167, and we observed that OTS167 did in fact impede cell proliferation at nanomolar concentrations (mean GI50 = 16 nM; see below). As OTS167 was able to inhibit growth despite the non-essentially of MELK, we considered the possibility that OTS167 acted through an off-target effect. To test this, we set out to determine whether MELK expression was actually required for OTS167 sensitivity. We calculated the GI50 value of OTS167 in A375, Cal51, and MDA-MB-231 cells that harbored gRNA's targeting either Rosa26 or MELK, and we found that cell populations with wild-type or mutant MELK displayed equivalent sensitivity to the drug (*Figure 3*). For instance, in Cal51 cells transduced with Rosa26 gRNAs, the GI50 values ranged from 9 nM to 12 nM (mean: 10 nM), while in Cal51 cells transduced with MELK gRNAs, the GI50 values ranged from 8 nM to 14 nM (mean: 11 nM). As OTS167 exhibits nanomolar potency against cancer cell lines but is unaffected by mutations in MELK, this suggests that OTS167 blocks proliferation by inhibiting another target or targets.

To further explore this observation, we tested the efficacy of OTS167 against a panel of triple-negative or receptor-positive breast cancer cell lines. MELK is significantly up-regulated in triple-negative tumors relative to receptor-positive tumors (*Wang et al., 2014*), and one clinical trial (NCT02926690) includes a dosage-escalation study of OTS167 in patients with triple-negative cancers. However, we observed no significant difference between the GI50 values of OTS167 according to receptor status (*Figure 3—figure supplement 1*). In triple-negative breast cancer cell lines, OTS167 inhibited growth by 50% at concentrations ranging from 10 nM to 42 nM (mean: 19 nM), while in receptor-positive breast cancer cells GI50 values ranged from 9 nM to 21 nM (mean: 14 nM). These results demonstrate that OTS167 is not specifically effective against triple-negative breast cancer cell lines, but instead remains remarkably potent against breast cancer cell lines that express hormone receptors.

Lastly, to confirm that OTS167 treatment fails to phenocopy MELK mutations, we examined their effects on cell cycle progression. We found that treatment with OTS167 blocked cytokinesis in a dose-dependent manner, resulting in populations harboring 13% to 60% multinucleate cells. In contrast, cells transduced with either Rosa26 or MELK gRNAs progressed through the cell cycle without gross mitotic defects, and exhibited no significant difference in the frequency of multinucleate cells

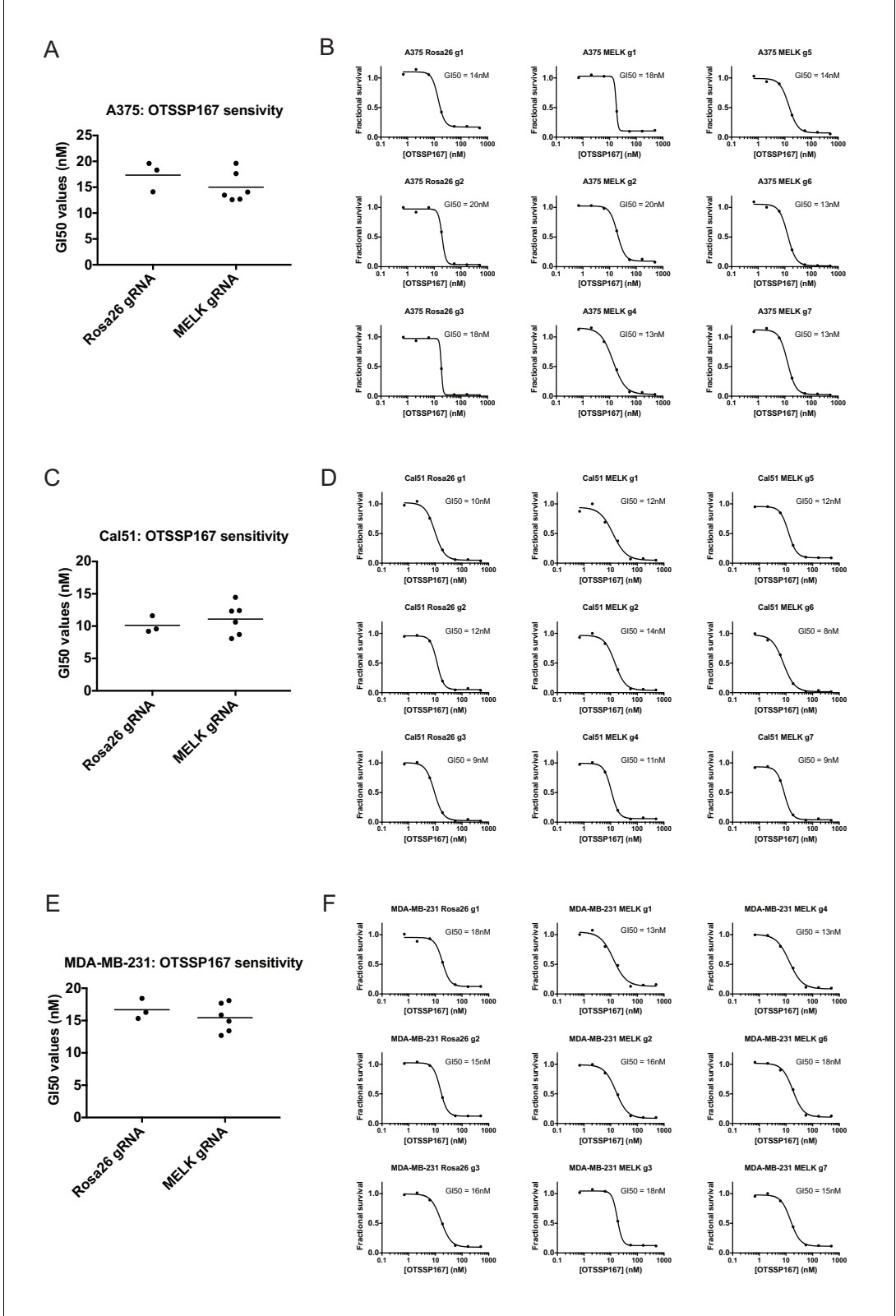

**Figure 3.** Mutating MELK does not affect OTS167 sensitivity. (**A**) Summary of GI50 values from OTS167 treatment of A375 cells harboring guide RNAs targeting Rosa26 or MELK. (**B**) 7 point dose-response curves of OTS167 in the indicated A375 cell lines. (**C**) Summary of GI50 values from OTS167 treatment of Cal51 cells harboring guide RNAs targeting Rosa26 or MELK. (**D**) 7 point dose-response curves of OTS167 in the indicated Cal51 cell lines.
*Figure 3 continued on next page*

*Figure 3 continued*

(**E**) Summary of GI50 values from OTS167 treatment of MDA-MB-231 cells harboring guide RNAs targeting Rosa26 or MELK. (**F**) 7 point dose-response curves of OTS167 in the indicated MDA-MB-231 cell lines.
The following figure supplements are available for figure 3:

**Figure supplement 1.** Receptor-positive breast cancer cell lines are sensitive to OTS167.
**Figure supplement 2.** OTS167 treatment, but not MELK mutation, causes the accumulation of multinucleate cells.

(*Figure 3—figure supplement 2*). These results demonstrate that OTS167 induces a cell cycle failure phenotype that is not recapitulated by mutagenizing MELK. We note that this observation is consistent with a recent publication that reported that, at certain concentrations, OTS167 was capable of inhibiting the mitotic kinases Aurora B, Haspin, and Bub1 (*Ji et al., 2016*). We conclude that the anti-proliferative effects of OTS167 are not a result of its inhibition of MELK.

## Generation and characterization of MELK-knockout clonal cell lines

To unambiguously demonstrate that MELK is dispensable for the proliferation of certain cancer cell lines, we used CRISPR to generate clonal cell lines that lack MELK protein. We transduced the MDA-MB-231 triple-negative breast cancer cell line with sets of 2 guide RNAs and then expanded clonal populations from single cells (*Figure 4A*). Recombination between the chosen gRNA cut sites eliminates exon 3, which encodes residues that are essential for ATP binding (*Cao et al., 2013*; *Cho et al., 2014*), as well as parts of exons 2, 4, and/or 5. We used PCR to identify three independent clones that were homozygous for CRISPR-induced recombination, and then confirmed the loss of the intervening genetic material by sequencing across the gRNA cut sites (*Figure 4B–C*). Additionally, we derived clones of Cal51 that have been transduced with single MELK gRNAs, and then identified three clones that harbored indels in the MELK kinase domain (*Figure 4—figure supplement 1*). Western blot analysis of the MDA-MB-231 and Cal51 clones with two antibodies that recognize distinct regions of MELK further verified the complete lack of MELK expression in all six derived cell lines (*Figure 4D–E* and *Figure 4—figure supplement 1*).

The MDA-MB-231and Cal51 MELK-KO clones exhibited robust proliferation, demonstrating that MELK is fully dispensable for the growth of these cancer cell lines (*Figure 4F* and *Figure 4—figure supplement 1*). In fact, one MDA-MB-231 MELK-KO clone exhibited a significantly shorter doubling time than the Rosa26 gRNA-transduced cell lines, potentially due to the presence of additional mutations that were acquired during clonal expansion. The MELK-KO clones progressed through the cell cycle without gross abnormalities and accumulated few multinucleate cells (*Figure 4G* and *Figure 4—figure supplement 1*). OTS167 treatment of MELK-KO clones caused the formation of multinucleate cells, demonstrating that this drug blocks cytokinesis by inhibiting another cellular target (*Figure 4H* and *Figure 4—figure supplement 1*). Finally, serial dilution analysis revealed that the MDA-MB-231 and Cal51 MELK-KO clones exhibited equivalent OTS167 GI50 values compared to Rosa26 gRNA-transduced lines (*Figure 4I–J* and *Figure 4—figure supplement 1*). We conclude that MELK is not an absolute requirement for triple-negative breast cancer proliferation, and that OTS167 blocks growth in a MELK-independent manner.

## Discussion

As a mitotic kinase highly expressed in many cancer types, MELK has been identified as a promising target for therapeutic intervention. However, through the use of CRISPR/Cas9-mediated mutagenesis, we have demonstrated that MELK is dispensable for growth in 13 out of 13 cancer cell lines tested, and that a MELK inhibitor currently in clinical trials blocks cell division by inhibiting another target. We believe that our results highlight the importance of using CRISPR/Cas9 technology to study and validate preclinical targets in cancer drug development.

Previous research utilizing RNA interference to knock down MELK has indicated that MELK expression is required for cancer cell proliferation. However, a growing body of evidence has revealed that RNAi is prone to pervasive off-target effects. This problem is particularly challenging

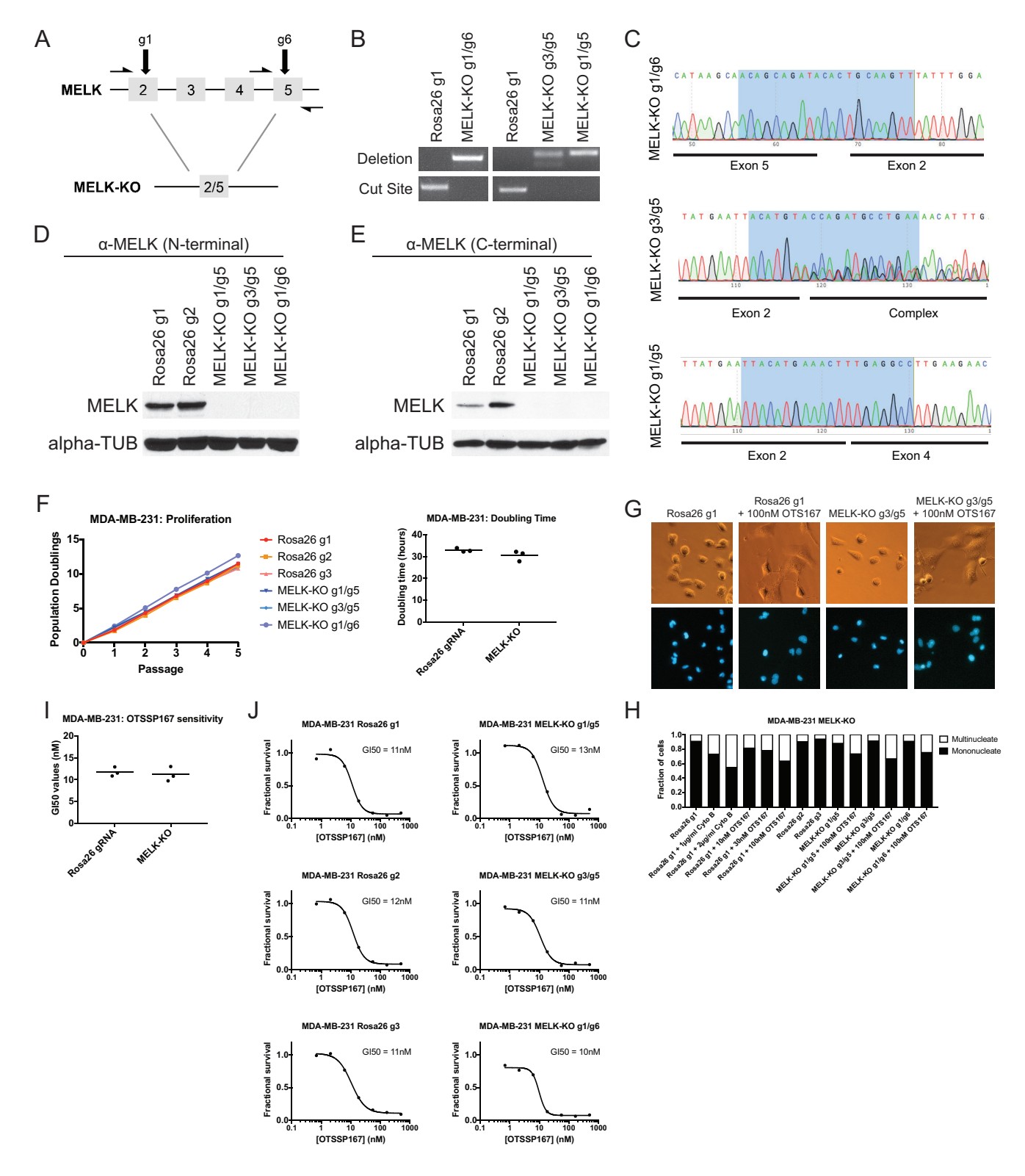

**Figure 4.** MELK-knockout cell lines proliferate at normal rates and remain sensitive to OTS167. (**A**) Schematic of exons in the MDA-MB-231 MELK-KO g1/g6 knockout line. Half-arrows indicate positions of either cut-site or deletion-spanning primers used to screen these colonies. Primer sequences are presented in ***Supplementary file 2***. (**B**) PCR validation of 3 independent MELK-KO clones. Note that amplification of the MELK-KO g3/g5 DNA with deletion-spanning primers yielded deletion products of at least two distinct sizes. (**C**) Sanger sequence validation of 3 independent MELK-KO clones.
*Figure 4 continued on next page*

*Figure 4 continued*

While MELK-KO g1/g6 and g1/g5 harbor a single homozygous deletion, MELK-KO g3/g5 harbors at least two distinct deletions. (D) Western blot analysis of MELK-KO clones using an antibody that recognizes a region in the N-terminal kinase domain (Abcam ab108529). (E) Western blot analysis of MELK-KO clones using an antibody that recognizes a region in the C-terminal domain (Cell Signal 2274S). (F) Proliferation analysis and doubling time measurements of MELK-KO cell lines. (G) Representative images of Rosa26 gRNA or MELK-KO clones either untreated or treated with 100 nM OTS167 and then stained with Hoechst dye. (H) The indicated cell lines were either left untreated or were treated with the cytokinesis inhibitor cytochalasin B or with OTS167. Cells were then stained with Hoechst dye. For each experiment, at least 200 cells were counted. (I) Summary of GI50 values from OTS167 treatment of either MDA-MB-231 Rosa26 gRNA or MELK-KO clones. (J) 7 point dose-response curves of OTS167 in the indicated cell lines.

The following figure supplement is available for figure 4:

**Figure supplement 1.** Generation and analysis of Cal51 MELK-KO cell lines.

when RNAi is used to study putative cell cycle regulators, as multiple publications have reported that the cell cycle genes *RAD51* and *MAD2* are unusually sensitive to off-target RNAi inhibition (*Adamson et al., 2012*; *Hübner et al., 2010*; *Sigoillot et al., 2012*). For instance, in a screen for genes whose depletion caused a bypass of the spindle assembly checkpoint, 34 of the top 34 candidate siRNA's exhibited off-target down-regulation of Mad2 levels (*Sigoillot et al., 2012*). Moreover, the expression of MELK is strongly cell-cycle regulated: MELK levels are typically low in G0/G1, and peak in mitosis [(*Badouel et al., 2010*) and our unpublished data]. A genetic or chemical treatment that induces a G1 arrest would therefore be predicted to down-regulate MELK, potentially confounding the analysis of knockdown efficiency. While Cas9 mutagenesis is also susceptible to off-target editing, to the best of our knowledge, the off-target loci affected by CRISPR are unlikely to substantially overlap with those that are affected by RNAi. Moreover, sequencing the locus targeted by Cas9 can provide an unbiased readout of mutagenesis efficiency that is not sensitive to cell state-dependent expression variability. Finally, unlike RNAi, CRISPR can be applied to generate clonal cell lines that harbor null mutations in a targeted gene. This technique bypasses the problems inherent in the analysis of mixed cell populations and partial loss-of-function phenotypes, and can provide significant insight into the genetic architecture of cancer.

One limitation of CRISPR mutagenesis is that, over the time required to generate or select for a pure cell population, cells may engage compensatory mechanisms to buffer against the loss of a targeted protein. Thus, the analysis of knockout clones can be complemented with cell-cell competition assays, which allow less time for cells to adapt to gene loss and may reveal the presence of a transient or immediate fitness defect induced by CRISPR. We performed a total of 91 competition assays (7 MELK gRNAs in 13 different cell lines) that failed to reveal an effect of MELK loss on cell fitness, further strengthening our conclusion that MELK is dispensable for cancer cell proliferation.

CRISPR mutagenesis can also assist in the pharmacological study of potential drugs. Several lines of evidence indicate that OTS167 does indeed inhibit MELK: for instance, a crystal structure of OTS167 binding to the MELK kinase domain has been reported (*Cho et al., 2014*). However, these structural and biochemical studies are unable to conclusively demonstrate that a phenotype in a living cell is due to an on-target effect. We believe that CRISPR represents a useful tool to gain genetic insight into this question: if a CRISPR-induced null mutation of a putative drug target fails to confer resistance to that drug, then that drug must act through alternate targets or mechanisms. While the MELK-KO cell lines that we generated remain exquisitely sensitive to OTS167, at present, we do not know how OTS167 blocks cell division. One possibility, not ruled out by our studies, is that OTS167 exhibits polypharmacology (*Knight et al., 2010*), and kills cancer cells by inhibiting multiple kinases, potentially including MELK. The analysis of drug-resistant alleles of other mitotic kinases that OTS167 has been shown to inhibit (*Ji et al., 2016*) may shed further light on the in vivo MOA of this compound.

Our results leave open the question of what role, if any, MELK plays in mammalian biology and cell cycle progression. While MELK is up-regulated in diverse tumor types, it is also expressed in several normal cell lineages, including embryonic cells, hematopoietic cells, and neural progenitor cells (*Heyer et al., 1997*; *Nakano et al., 2005*; *Gil et al., 1997*). MELK may be required at a certain developmental stage, or for a specific cell type or organismal process. Similarly, we cannot currently rule out the possibility that MELK plays a role in tumorigenesis in vivo that was not assessed in our

current work. At a minimum, our results suggest that MELK is dispensable for mitotic progression in most cancers. MELK may function in an overlapping or redundant pathway with other mitotic kinases, several of which are up-regulated along with MELK in tumor cells (*Malumbres and Barbacid, 2007*). Synthetic lethal screens and further in vivo investigation will shed light on MELK's function in development and cancer. Nonetheless, our data suggest that specific MELK inhibitors are unlikely to be useful monoagents in cancer therapy.

## Materials and methods

### Tissue culture

The identity of every human cell line utilized in this paper was authenticated using STR profiling (University of Arizona Genetics Core, Tucson, AZ). Cell lines were also confirmed to be negative for mycoplasma contamination using the MycoAlert Detection Kit (Lonza, Switzerland; LT07-218). Cell lines HCC70, HCC1937, MDA-MB-453, MDA-MB-468, and ZR-75–1 were grown in RPMI 1640 supplemented with 10% fetal bovine (FBS), 2 mM glutamine, 1% Nonessential Amino Acids (Life Technologies, Waltham, MA; BY00148), and 100 U/ml penicillin and streptomycin. HCC1143 and NCI-H1299 were grown in RPMI 1640 supplemented with 10% FBS, 2 mM glutamine, and 100 U/ml penicillin and streptomycin. T24 was grown in McCoy's 5A media supplemented with 10% FBS, 2 mM glutamine, and 100 U/ml penicillin and streptomycin. Cal51, A375, MDA-MB-231, HCT116, Cama1, JIMT-1, and U118-MG cells were grown in DMEM supplemented with 10% FBS, 2 mM glutamine, and 100 U/ml penicillin and streptomycin. T47D cells were grown in RPMI supplemented with 10% FBS, 6.94 µg/ml insulin (Thermo Fisher, Waltham, MA; BN00226), 2 mM glutamine, and 100 U/ml penicillin and streptomycin. MCF7 cells were grown in DMEM supplemented with 10% FBS, 0.01 mg/ml insulin, 2 mM glutamine, and 100 U/ml penicillin and streptomycin. Cell lines were kindly provided by the individuals thanked in the acknowledgments. All cell lines were maintained in a humidified environment at 37°C and 5% $CO_2$. Cell counting was performed using the Cellometer Auto T4 system (Nexcelom, Lawrence, MA).

### Plasmid construction

Guide RNAs targeting protein domains in MELK, PCNA, and RPA3 were designed with assistance from Osama El Demerdash (manuscript in preparation). Oligonucleotides were ordered from IDT and then cloned into the LRG 2.1 vector [a gift from Jun-Wei Shi (University of Pennsylvania) and Chris Vakoc (Cold Spring Harbor Laboratory)] using a BsmBI digestion (*Shalem et al., 2014*). Plasmids were amplified in Stbl3 E. coli (Thermo Fisher; C737303) prepared using the Mix and Go transformation kit (Zymo Research, Irvine, CA; T3001).

### Plasmid transfection and transduction

HEK293T cells were transfected using the calcium-phosphate method (*Smale, 2010*). Supernatant was harvested 48 to 72 hr post-transfection, filtered through a 0.45 µm syringe, and then frozen at −80° C for later use or applied directly to cells with 4 µg/mL polybrene. The culture media on target plates was changed 24 hr post-transduction.

### Proliferation analysis

To measure cell proliferation, 100,000 cells of each strain were plated on a six well plate and then allowed to grow for 72 hr. Cells were then trypsinized, counted, and 100,000 cells were re-plated in fresh media. Cells were passaged five times, and cumulative population doublings and doubling times were calculated at each passage.

### Soft agar assays

Solutions of 1.0% and 0.7% Difco Agar Noble in sterile water were autoclaved and then allowed to equilibrate at 42°C or 37.5°C, respectively. The 1% agar solution was then mixed 1:1 with 2 ml of the appropriate media, supplemented with 2X the concentration of serum, glutamine, and penicillin/ streptomycin. 1 mL of this mixture was then plated in one well of a six well plate and allowed to solidify at room temperature for 30 min, resulting in a 0.5% agar base layer. Cells of interest were then trypsinized and re-suspended in their appropriate media. 20,000 cells were diluted in 1 ml of

2X media and then mixed with 1 ml of the 0.7% agar solution. 500 µl of this mixture was then plated on the solidified base layer, resulting in 5000 cells in a 0.375% agar suspension. The agar was allowed to solidify at room temperature for one hour before being transferred to an incubator. Fresh 1X media was added to the surface of each well 24 hr after plating, and then were re-fed every 3 days. After 21 days of growth, colonies were scored under 10x magnification. All experiments were plated in triplicate and performed twice.

## Western blot analysis

Cells were lysed with RIPA buffer [25 mM Tris, pH 7.4, 150 mM NaCl, 1% Triton X 100, 0.5% sodium deoxycholate, 0.1% sodium dodecyl sulfate, protease inhibitor cocktail (Roche, Indianapolis, Indiana), phosphatase inhibitor cocktail (Roche)]. Lysates were quantified using the Pierce BCA Kit (Thermo Scientific), and equal amounts of protein were denatured and loaded onto an 8% SDS-PAGE gel. The protein was transferred onto a polyvinylidene difluoride membrane using the Trans-Blot Turbo Transfer System (Bio-Rad, Hercules, California). The membrane was blocked in 5% non-fat milk-TBST and then incubated with anti-MELK (abcam, Cambridge, MA; ab108529) at a 1:3000 dilution, or blocked in 10% BSA-TBST and then incubated with anti-MELK (Cell Signal, Danvers, MA; 2274S) at a 1:1000 dilution. Anti-alpha-tubulin (Sigma-Aldrich, St. Louis, MO; T6199) was used as a loading control at a 1:3000 dilution. All primary antibody incubations were performed overnight at 4°C. Following incubation, the membranes were washed and then incubated in either anti-rabbit secondary (abcam; ab6721) at 1:50000 for MELK or anti-mouse secondary (Bio-Rad; 1706516) at 1:10000 for Tubulin for 1 hr at room temperature.

## Analysis of CRISPR-mediated mutagenesis

Genomic DNA was extracted from transduced cell lines using the QIAmp DNA Mini kit (Qiagen, Germantown, MD; Cat. No. 51304). Loci targeted by guide RNAs were amplified using the primers listed in *Supplementary file 2*, and then sequenced using the forward and reverse primers at the Cold Spring Harbor Laboratory sequencing facility. Sequence traces were analyzed using TIDE (*Brinkman et al., 2014*).

## Analysis of OTSSP167 sensitivity

For every cell line of interest, 10,000 cells were plated in 100 µl of media in an 8 × 4 matrix on a flat-bottomed 96-well plate. Cells were allowed to attach for 24 hr, at which point the media in every well was changed. 500 nM of OTSSP167 (MedChem Express, Monmouth Junction, NJ; Cat. No. HY-15512A) was added to one row of cells, and then 6 3-fold serial dilutions were performed. After 72 hr of growth in the presence of the drug, cells were trypsinized and counted using a MacsQuant Analyzer 10 (Milltenyi Biotec, Germany). The fraction of cells recovered at every drug concentration, relative to a row of untreated cells, was determined. GI50 values were calculated using a four-parameter inhibition vs. concentration model in Prism 7 (Graphpad, San Diego, California). Sensitivity experiments in *Figure 4J* and *Figure 3—figure supplement 1* were performed 2–3 times each, while sensitivity experiments in *Figure 3* and *Figure 4—figure supplement 1* were performed once.

## GFP dropout screening

Cells were transduced on day 0 with sgRNA lentiviral supernatant, which was then replaced with fresh media on day 1. On day 3, the baseline percentage of GFP+ cells was measured using a MacsQuant Analyzer 10 (Milltenyi Biotec). Cells were then passaged every 3 or 4 days, according to their growth rate and confluence, and the percentage of GFP+ cells was measured at every split. Dropout values represent the fold decrease in GFP+ cells at each passage, relative to the GFP+ percentage on day 3. In preliminary experiments with A375 and MDA-MB-231, replicate dropout assays were highly reproducible across independent replicates. For that reason, GFP dropout experiments in the 13 tested cell lines were performed once.

## DNA staining

10,000 (A375, Cal51) or 20,000 (MDA-MB-231) cells of interest were plated in 250 µl of media in a flat-bottomed 24-well plate and allowed to attach for 24 hr. Then the media was replaced, and OTS167 or Cytochalasin B (Cayman Chemical Company, Ann Arbor, MI; Cat. No. 11328) were added

to control wells. Following an additional 24 hr period of growth, cells were stained with 2.5 μg/ml of Hoechst dye (Thermo Fisher, Cat. No. H3569) for 30 min and imaged using appropriate filters. DNA staining experiments were performed twice.

## Acknowledgements

We thank Chris Vakoc and Junwei Shi for providing Cas9 and guide RNA expression plasmids. We thank Chris Vakoc, Camila dos Santos, and Ken Chang for providing cell lines. We thank Allison Leonard for technical assistance with lentiviral production. We thank Osama El Demerdash for assistance designing domain-targeted guide RNAs. This work was performed with assistance from CSHL Shared Resources, including the CSHL Flow Cytometry Shared Resource, which are supported by the Cancer Center Support Grant 5P30CA045508. Research in the Sheltzer Lab is supported by an NIH Early Independence Award (1DP5OD021385).

## Additional information

### Funding

| Funder | Grant reference number | Author |
| --- | --- | --- |
| National Institutes of Health | 1DP5OD021385 | Jason M Sheltzer |

The funders had no role in study design, data collection and interpretation, or the decision to submit the work for publication.

### Author contributions

AL, CJG, NMS, Investigation, Methodology; JMS, Conceptualization, Supervision, Funding acquisition, Investigation, Methodology, Writing—original draft, Writing—review and editing

### Author ORCIDs

Ann Lin, http://orcid.org/0000-0002-4618-8120

Christopher J Giuliano, http://orcid.org/0000-0002-0586-6095

Nicole M Sayles, http://orcid.org/0000-0001-7460-9095

Jason M Sheltzer, http://orcid.org/0000-0003-1381-1323

## Additional files

### Supplementary files

• Supplementary file 1. Guide RNA sequences. The sequences of every guide RNA and the protein domain targeted by the guide RNAs are displayed.

• Supplementary file 2. PCR primers to amplify MELK gRNA cut sites and deletions. The sequences of PCR primers used in this manuscript are displayed.

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
