## [Decision Letter]

Thank you for submitting your article "CRISPR/Cas9 mutagenesis invalidates a genetic target of clinical trials in cancer" for consideration by *eLife*. Your article has been favorably evaluated by Tony Hunter (Senior Editor) and two reviewers, one of whom is a member of our Board of Reviewing Editors. The reviewers have opted to remain anonymous.

The reviewers have discussed the reviews with one another and the Reviewing Editor has drafted this decision to help you prepare a revised submission.

Summary:

Lin and colleagues report their CRISPR-based analysis of the candidate oncology drug target, MELK, a kinase that has been previously implicated in cancer in numerous publications, and for which a pharmacologic inhibitor is currently being evaluated in early clinical trials. There has been particular interest in its role in triple-negative breast cancer. In short, they made use of CRISPR cutting technology to generate MELK mutant alleles in a panel of cancer cell lines, including several in which MELK dependency has previously been reported. In cell culture assays including competitive cell growth, proliferative potential, and anchorage-independent growth, they used these cells to demonstrate that, in contrast to previous claims, MELK disruption did not seem to significantly affect the oncogenic potential of any of the tested cell lines. Furthermore, after generating single cell-derived MELK-deficient clones of one of these lines, they showed that the clinical small molecule MELK inhibitor retained its growth inhibitory properties, suggesting that this inhibitor displays off-target activities that are responsible for the observed effects on cell growth. They also examined published datasets associated with independently performed CRISPR and RNAi screens that included MELK-targeted agents and concluded that MELK dependency was not observed in any of those unbiased screens. Based on these findings, the authors conclude that previously reported MELK dependency in cancer is likely to reflect an experimental artifact, and that the MELK inhibitor that is currently being evaluated in the clinic may be functioning through a distinct, currently unknown mechanism.

Overall, this is a well-executed and well documented study. Although these are essentially "negative results", the fact that others are still pursuing MELK as a candidate therapeutic target, and that clinical trials focused on MELK inhibition are underway, justifies publication of the findings.

Essential revisions:

1) The authors need to demonstrate that the MELK protein is substantially depleted in cell lines following CRISPR engineering. They have shown this for one cell line by western blot, but it should be shown for a few of the other lines as well to rule out the possibility that in some (or even many) of these lines, a small amount of "residual", non-mutated, expressed MELK is sufficient to maintain normal proliferation.

2) The authors have not mentioned one of the most likely possibilities for the MOA of OTS167, which is polypharmacology. It could even be that OTS167 inhibition of MELK + Target X explains how it kills cells. Most kinase inhibitors, including the prominent approved drugs, Gleevec (bcr-abl + ckit), dasatinib (Bcr-abl + src family kinases), Sutent (many) work in this manner. The single gene knockout studies here would not show differences in this case. This should be acknowledged explicitly.

3) The authors should also address other possible explanations of the difference between genetic perturbation (compensation could take place by upregulation of other kinases during the time it takes to deplete the protein after gene editing) and pharmacology (instantaneous inhibition, which might not give time for the cells to upregulate compensating systems). These differences can be acute in the setting of oncogene addiction/cancer cell signaling.

---

## [Author Response]

*Essential revisions:*

*1) The authors need to demonstrate that the MELK protein is substantially depleted in cell lines following CRISPR engineering. They have shown this for one cell line by western blot, but it should be shown for a few of the other lines as well to rule out the possibility that in some (or even many) of these lines, a small amount of "residual", non-mutated, expressed MELK is sufficient to maintain normal proliferation.*

To address the reviewers’ concern that residual MELK protein may remain in our cell lines, we derived a second set of MELK-knockout clones, this time from the Cal51 triple-negative breast cancer cell line (Figure 4—figure supplement 1). We confirmed the presence of large indel mutations in the MELK kinase domain by Sanger sequencing, and verified the complete loss of MELK protein expression using two antibodies that recognize distinct epitopes. Our manuscript now includes six MELK-null clones from two different cell lines (MDA-MB-231 and Cal51), none of which show a fitness or cell cycle progression defect in any experiment that we have thus far performed.

We also conducted additional western blot analyses on sorted populations of A375 and Cal51 cells transduced with MELK gRNAs (Figure 1—figure supplement 4). We observed strong depletion of MELK in all cell populations tested.

In total, our manuscript presents functional evidence that our CRISPR system is active (dropout of gRNAs targeting PCNA and RPA3), sequencing evidence that our MELK gRNAs efficiently induce indel mutations in the MELK gene, immunoblot evidence that these indel mutations result in the loss of MELK protein expression in multiple independent cell lines, and we confirm that no residual MELK protein is left in six MELK-knockout clonal lines.

*2) The authors have not mentioned one of the most likely possibilities for the MOA of OTS167, which is polypharmacology. It could even be that OTS167 inhibition of MELK + Target X explains how it kills cells. Most kinase inhibitors, including the prominent approved drugs, Gleevec (bcr-abl + ckit), dasatinib (Bcr-abl + src family kinases), Sutent (many) work in this manner. The single gene knockout studies here would not show differences in this case. This should be acknowledged explicitly.*

In light of the apparent promiscuity of OTS167 (Ji et al., PLOS One, 2016), we agree with the reviewers that polypharmacology is a likely MOA of OTS167. We have added a discussion of this possibility in the fourth paragraph of the Discussion section.

*3) The authors should also address other possible explanations of the difference between genetic perturbation (compensation could take place by upregulation of other kinases during the time it takes to deplete the protein after gene editing) and pharmacology (instantaneous inhibition, which might not give time for the cells to upregulate compensating systems). These differences can be acute in the setting of oncogene addiction/cancer cell signaling.*

We agree that one potential drawback of CRISPR is the lag time between the introduction of a gRNA, the loss of the targeted protein, and the ability to analyze a resulting phenotype. In particular, during the periods of growth required to generate and analyze pure GFP+ populations or MELK-knockout clones, it is possible that cells could “adapt” to the loss of MELK. However, we believe that the cell competition assays that we also performed give us insight into the more immediate consequences of MELK loss, as cells are under observation from the time in which the gRNA is introduced onward. A transient loss of cell fitness, followed by adaptation, would be detectable as a decrease in GFP+ cells in passage 2 or 3 of this assay, followed by stabilization of the GFP+/GFP- ratio. In total, we conducted 91 independent cell competition assays (7 MELK gRNAs in 13 cell lines) that could have revealed a transient fitness defect, but we failed to detect any evidence that this was occurring. It’s true that we cannot rule out the possibility that immediate adaptation is taking place, e.g. cells are able to compensate for MELK loss in such a way that not even a transient fitness defect is evident, but we believe that our experiments have given us a good chance to detect anything short of perfect compensation. To clarify these points, we have added a discussion of this issue in our manuscript (subsection "Mutagenizing MELK using CRISPR/Cas9” and Discussion, third paragraph).